# Long-Term Efficacy and Safety of Ibrutinib in the Treatment of CLL Patients: A Real Life Experience

**DOI:** 10.3390/jcm10245845

**Published:** 2021-12-13

**Authors:** Alessandro Broccoli, Lisa Argnani, Alice Morigi, Laura Nanni, Beatrice Casadei, Cinzia Pellegrini, Vittorio Stefoni, Pier Luigi Zinzani

**Affiliations:** 1IRCCS Azienda Ospedaliero-Universitaria di Bologna, Istituto di Ematologia “Seràgnoli”, 40138 Bologna, Italy; alessandro.broccoli@studio.unibo.it (A.B.); alice.morigi2@studio.unibo.it (A.M.); laura.nan@libero.it (L.N.); bea.casadei@gmail.com (B.C.); cinzia.pellegrini5@unibo.it (C.P.); vittorio.stefoni2@unibo.it (V.S.); 2Dipartimento di Medicina Specialistica, Diagnostica e Sperimentale, Università di Bologna, 40138 Bologna, Italy; lisa.argnani@unibo.it

**Keywords:** chronic lymphocytic leukemia, ibrutinib, Richter transformation

## Abstract

Ibrutinib has demonstrated a significant clinical impact in patients with de novo and relapsed/refractory chronic lymphocytic leukemia (CLL), even in cases with unfavorable cytogenetics and molecular markers. All CLL patients’ data treated at our Institute with ibrutinib have been retrospectively reviewed. Forty-six patients received ibrutinib either as frontline (10) or second or more advanced treatment (36). Five patients presented with TP53 mutations; 11 had the deletion of chromosome 17p; 17 displayed an unmutated immunoglobulin variable heavy chain status. The median number of cycles administered was 26. Among patients treated frontline, the best overall response rate (ORR) was 90.0%. In patients receiving ibrutinib as a second or later line ORR was 97.2%. Median progression-free survival was 28.8 and 21.1 months for patients treated frontline and as second/later line, respectively. Median overall survival was not reached for those treated frontline and resulted in 4.9 years for patients treated as second/later line. Grade 3–4 hematological toxicities were neutropenia, thrombocytopenia, and anemia. Grade 3–4 extrahematological toxicities included diarrhea, cutaneous rash, utero-vesical prolapse, vasculitis, and sepsis. Ibrutinib is effective and well tolerated in CLL. Responses obtained in a real-life setting are durable and the safety profile of the drug is favorable.

## 1. Introduction

Ibrutinib is a first-in-class covalent Bruton tyrosine kinase (BTK) inhibitor that blocks the B-cell receptor (BCR) signaling within chronic lymphocytic leukemia (CLL) cells, a mechanism that plays a critical role in initiating and maintaining the disease, as well as in contributing to its progression [1,2]. Given orally as a single agent at the initial dose of 420 mg/day, ibrutinib has demonstrated its greatest effectiveness in symptomatic CLL patients, either as a salvage treatment in those failing a previous approach with chemo-immunotherapy, or as a frontline strategy, as demonstrated in the pivotal phase 1b/2 PCYC-1102 trial [3,4].

Updated follow-up analyses of the registration trials in relapsed and refractory patients have shown extensive benefits in terms of progression-free survival (PFS) and overall survival (OS) [3,4], especially when ibrutinib was compared with ofatumumab [5,6], the best available option for the treatment of patients failing previous systemic approaches, in the phase 3 RESONATE trial. Significant results have also been shown in patients treated frontline, specifically in patients with advanced age and comorbidities [3,4,7], and a clear superiority of ibrutinib over chlorambucil alone was demonstrated in the phase 3 RESONATE-2 study. Moreover, ibrutinib seems capable to overcome the negative impact conferred by high-risk genomic features, such as the deletion of the short arm of chromosome 17 [del(17p)] and the mutation of TP53 (TP53mut) [3,4,7,8,9,10], and to confer benefits in cases with an unmutated immunoglobulin variable heavy chain (IGHVunmut) status [11], which is a known negative predictive factor for long-term survival in patients treated frontline with chemoimmunotherapy [12].

Despite the undisputed benefit on survival functions and the significant incidence of responses, the amount of complete responses (CR) remains low [3,4,5,6,7]: this means that a continuous administration of the drug is required to keep patients in remission, as a tonic inhibition of BTK is necessary to counteract the continuous regeneration of fresh BTK protein in neoplastic cells [13].

Apart from the information collected in clinical trials, there is scarcity of data on the use of ibrutinib in real-life settings with patient series which were presumably not qualified for clinical trials. More precisely, the existing experience with patients outside of trials confirms the efficacy of ibrutinib in terms of response, but only a few studies have mature data on their long-term outcomes, survival, and adherence to treatment [14,15,16,17]. To date, the long-term toxicity profile of ibrutinib is well characterized and includes a clinically significant incidence of cardiac arrhythmias, bleeding, infection, diarrhea, arthralgias, and hypertension [14,15,16,17].

The aim of the present study was to report a single-centre real-life experience with ibrutinib in CLL patients, according to the Italian prescribing rules which allow the administration of this drug in the relapsed and refractory setting, regardless of any treatment received previously, and in the frontline setting in case of adverse cytogenetic features (del(17p)), TP53mut, or in patients with more than 65 years.

## 2. Materials and Methods

### 2.1. Study Overall Conduct

A single-center observational retrospective study was conducted on patients affected by CLL followed at our institution, in need of treatment, and considered eligible for ibrutinib therapy. The study was performed in accordance with the ethical standards as laid down in the 1964 Declaration of Helsinki and its amendments. Patients were consecutively considered to avoid selection bias.

The diagnosis of CLL had to be established by peripheral blood flow cytometry. Patients were included in the study either if they required frontline intervention or if displayed symptomatic relapsed or refractory disease. We established a minimum treatment duration with ibrutinib of at least 12 months in order to confirm eligibility. Fluorescence in situ hybridization (FISH) for del(17p) and molecular evaluation of IGHV and TP53 mutational status were considered essential requirements before starting any frontline therapy. FISH and TP53 mutational status have been repeated before ibrutinib inception in all patients. IGHV mutational status was not generally evaluated more than once. Cytogenetic analysis was performed in all patients but 11q is not routinely assessed. According to the recently updated International Workshop on CLL recommendations, thorough genetic risk stratification in CLL requires FISH analysis complemented by mutational screening for the TP53, IGHV, and del(17p) [18].

Oral ibrutinib was administered at the conventional dose of 420 mg/day, continuously and up to disease progression or unacceptable intolerance. Dose delays and modifications have been made when appropriate according to the summary of product characteristics. Patients were clinically evaluated once a month and their hematology and biochemical values collected at each visit. Imaging procedures (computed tomography scan of neck, thorax and abdomen or abdominal ultrasonography, as considered appropriate) were performed as per institutional guidelines before ibrutinib initiation, after the completion of the 4th (±1) month of treatment (i.e., initial or interim response), then every six months as per institutional procedures. Imaging could be anticipated in case of suspect clinical progression. Bone marrow biopsy was required to confirm CR status.

### 2.2. Study Endpoints

This retrospective analysis was intended to provide details on effectiveness, survival, and tolerability of ibrutinib-treated CLL patients at a single institution. Overall response rate (ORR), CR rate, PFS, and OS were the principal study endpoints. Time-to-next treatment (TNT) was also estimated. Responses have been categorized as per International Workshop on CLL criteria [18]. Best response was considered as the most favorable disease status achieved at any point during treatment. Safety and tolerability of the treatment were assessed by recording type, incidence, and severity of any adverse events (AEs) in accordance with the National Cancer Institute Common Terminology Criteria for Adverse Events (version 4.0).

### 2.3. Statistical Analysis

Demographics and patients’ characteristics were summarized by descriptive statistics and time-to-point events were estimated by using the Kaplan–Meier method and compared using log-rank test. Statistical analyses were performed with Stata 11 (StataCorp LP, College Station, TX, USA) and *p*-values were set at 0.05.

## 3. Results

### 3.1. Patients

Forty-six patients received ibrutinib between February 2016 and October 2019 either as frontline (10 patients) or second or more advanced treatment in case of disease refractoriness or relapse (36 patients). All patients reached a time-on-treatment of at least 12 months.

Median age at CLL diagnosis was 62 (range 33–79) years. Thirty-one patients were males and 15 females. Overall, 5 patients had TP53mut at treatment inception, 11 displayed del(17p), and 17 had an IGHVunmut status. Clinical, molecular, and cytogenetic characteristics for each group of treated patients (frontline versus 2nd or more advanced line) are summarized in Table 1.

Pretreated patients received a median of one previous treatments (range 1–4), which consisted of chemoimmunotherapy (rituximab + bendamustine or fludarabine combinations) in 81% of cases. The latest treatment combination before ibrutinib was represented by chemoimmunotherapy in 30 patients (83%), idelalisib + rituximab in 4 patients (11%), umbralisib and single-agent anti-CD19 therapy in one patient each (3%). Thirteen patients (36%) were refractory to the last treatment they have received immediately before ibrutinib. Patients who received ibrutinib frontline did so because of unfavorable cytogenetic or molecular status at disease onset or because of their unfitness to chemoimmunotherapy due to their age or comorbidity.

### 3.2. Response to Treatment and Survival Analysis

Among patients treated frontline, best responses included one CR and 9 partial responses (PR), yielding an ORR of 100%. Among patients receiving ibrutinib as a second or later line, best responses were one CR and 34 PR, with an ORR of 97.2%. Figure 1 represents the incidence of CR and PR at early (interim) evaluation up to the best achieved response in both subsets of patients.

At a median follow-up of 24 and 26 months for patients treated frontline and as a second or later line, respectively, median PFS were 28.8 and 21.1 months for each subgroup. Median OS was not reached for those treated frontline and it was 4.9 years for patients receiving ibrutinib as salvage therapy. Outcomes according to treatment line are summarized in Table 2 and reported in Figure 2.

PFS at 1, 2, and 3 years for patients receiving ibrutinib as a salvage treatment was 87.2%, 44.3%, and 22.1%, respectively, while OS at the same time points was 100%, 92.3%, and 68.8%. Both survivals did not statistically differ (*p* = 0.187 and 0.08, respectively).

Richter transformation (RT) occurred in 5 patients (11%) at a median time of 16 months since the initial dose. At a median time of 3.8 years, 12 patients required further therapy: 10 patients shifted to venetoclax and 2 to chemotherapy because of RT. TNT curves are shown in Figure 3 (no statistical difference occurred between the two cohorts with *p* = 0.344).

### 3.3. Treatment Administration

The median number of cycles administered was 26, ranging from 12 to 80. All patients started with the standard dose of 420 mg/day. A dose reduction was performed in 14 patients (30%) due to hematologic (8 patients, 17%) and extra-hematologic toxicity (6 patients, 13%; see below for full details). The drug was temporarily held in 20 instances, due to AES in 15 cases and to surgical procedure or invasive interventions in 5 cases.

At the time of writing, 31 patients (67%) have discontinued treatment, with a median time on treatment of 26 (range 12–81) months. The most frequent cause for discontinuation was progressive disease (PD), which occurred in 27 cases (59%). Causes of early discontinuation other than PD were represented by sepsis (2 patients), hepatitis B virus reactivation (1 patient), and cutaneous toxicity (1 patient). Among the 15 patients who are still receiving ibrutinib (33%, 3 as a frontline treatment and 12 as a second or more advanced line), the median time on treatment is 22 (range 15–45) months.

### 3.4. Safety

Overall, 24 patients displayed at least one toxicity. Seventeen hematological AEs were documented in 13 patients, consisting of 10 episodes of neutropenia (6 grade 4 and 4 grade 3 episodes), 3 of anemia (grade 4 in 1 case and grade 3 in 2 cases), and 4 of thrombocytopenia (grade 4 in 2 cases and grade 1–2 in 2 cases). One grade 1 episode of thrombocytopenia was due to an autoimmune mechanism and considered unrelated to ibrutinib. In eight cases, AEs determined a dose reduction of the drug, while in one case the next dose was only temporarily withhold. In the remaining eight cases, patients with cytopenia recovered without modification of the administration schedule and with concomitant medications only. One patient displayed persistent bilinear cytopenia, consisting of grade 4 thrombocytopenia and grade 3 neutropenia (Table 3).

Nineteen patients displayed at least one extrahematological toxic effect, with an overall incidence of 43 AEs. Among those, the most clinically meaningful were represented by one grade 4 sepsis (secondary to urinary tract infection) and 4 grade 3 events (vasculitis, uterovesical prolapse with concomitant hydronephrosis, cutaneous rash and diarrhea). All extrahematological AEs have recovered, irrespective of their severity: in 6 cases, a dose reduction of ibrutinib was required (grade 1 and grade 2 fever, grade 2 joint pain in 2 cases, grade 3 vasculitis, grade 3 diarrhea).

Ten severe AEs were recorded in 9 patients; 5 of them were considered correlated with ibrutinib (1 case of pneumonia and 4 cases of sepsis), while the remaining were unrelated (COVID-19 infection, acute psychosis, acute renal failure, and chronic obstructive pulmonary disease exacerbation).

## 4. Discussion

Treatment with ibrutinib nowadays represents the mainstay of the CLL management: clinical trials have demonstrated efficacy in the relapsed and refractory setting, with higher efficacy when given as first salvage treatment rather than later in the course of the disease [4], with an undisputable benefit when compared to other agents active in the same setting [6]. Moreover, it represents the first choice of frontline treatment in CLL patients with high-risk cytogenetic and molecular features, and it is also capable of abrogating the prognostic gap that exists between chemoimmunotherapy-treated patients bearing, respectively, IGHVunmut and mutated genes [4,9,10,11]. Recently, single-agent ibrutinib has also challenged the standard frontline chemoimmunotherapy paradigm both in young and fit patients and in elderly and less fit individual with no del(17p) nor TP53mut [19,20]: in this sense, it appears extremely versatile and applicable in any context of therapy.

The present study shows the effectiveness of ibrutinib in a real-life experience with 46 patients, either receiving the drug as first-line therapy or as a salvage treatment, in accordance with the current Italian prescription rules. We have confirmed high response rates both in the frontline and in the pre-treated setting (100.0% and 97.2%, respectively), although with a limited number of CR (10.0% and 2.8%, respectively), as previously demonstrated in registration trials, and in line with previously published real-world experiences with relapsed and refractory patients (Table 4) [14,15,16,17].

Given the important limitation of the number of patients in each treatment setting in our series, it is however worth noting that our population treated frontline with ibrutinib displays more adverse features in comparison to patients in registration trials: del(17p) was found in 60% in our cohort of 10 patients, while the incidence was 6% in the PCYC-1102 trial, and no del(17p) were enrolled in the phase 3 RESONATE-2 trial, as it was an exclusion criterion [4,7]. Moreover, our patients had TP53mut in 40% of cases and an IGHVunmut status in 50%. This difference in terms of enrolment characteristics may in part explain the shorter PFS and OS we have observed in our frontline-treated cases. In addition, we reported a monocentric experience of a referral hospital in which more than 200 trials are ongoing: this could explain a shorter TNT, as we have more available new drugs. Response rates appear instead rather similar across studies (Table 5).

Likewise, discrepancies observed between patients treated in second or later line in our study and in clinical trials may be in part explained by the fact that possible more comorbidities affecting patients treated on a routine basis may have affected the compliance to therapy, the emergence of AEs, and ultimately, the outcome of the treatment itself (Table 4) as previously reported [21,22]. Therefore, if on the one hand, it is important to operate a comparison between real-life data and clinical trial outcomes, on the other, the information gathered in real-world experiences is undoubtedly valuable to help clinicians acquire familiarity with the use of new agents in several contexts of treatment.

Reports from real-life studies are also important to outline the safety profile and the manageability of a drug. Our experience confirms that ibrutinib is safe and that emergent toxicities can be easily managed with concomitant medications or dosing reductions, resulting in hospitalizations only in a limited proportion of cases (10 severe AEs out of 60 AEs in our population) and just occasionally in treatment interruptions. Importantly, we have not reported any event of atrial fibrillation nor bleeding in our population, both regarded as events of special interest in case of treatment with ibrutinib [23,24]. Moreover, the incidence of infectious complications and severe infectious AEs was rather low (one case of pneumonia and 4 sepsis, all recovered), overall representing 50% of the severe AEs observed and only 8% of any incident AE. Unfortunately, there are scarce data about the occurrence of Richter’s syndrome in this setting even if we acknowledge that 11% may be considered a high percentage. The familiarity gained over a period of more than five years has helped to manage AEs adequately and has taught physicians to establish pre-treatment screening procedures and to refine monitoring strategies to prevent the onset of potentially harmful side effects (e.g., bleeding) or toxicities that can jeopardize patients’ adherence to treatment. Cooperation with diverse clinical specialties seems the key to success for the optimal management of ibrutinib-treated patients [24,25].

The importance of a continuous administration of ibrutinib to maintain a status of remission is counteracted by the adequate compliance to treatment, which is a major concern in case of elderly and severely comorbid patients. Indefinite treatment paradigms, as the ones represented by ibrutinib itself and by the single-agent BCL2-inhibitor venetoclax [26], are nowadays challenged by fixed term combinations at CLL relapse and by minimal residual disease-driven definite term strategies in the next future [27,28]. The identification of patients who may take advantage of continuous treatments rather than fixed term therapies is hard, as no head-to-head comparisons are available between BCR and BCL2 inhibitors to date. Patients’ age, comorbidities, safety profile, compliance, and life expectancy are all relevant factors to be taken into account to operate the best choice between these alternatives.

Of note, the forthcoming advent in Italian everyday clinical practice of second generation BTK inhibitors (e.g., acalabrutinib, zanubrutinib), which have demonstrated comparable efficacy, but are expected to have fewer AEs than ibrutinib, might yet again change the treatment paradigm in CLL.

## Figures and Tables

**Figure 1 jcm-10-05845-f001:**
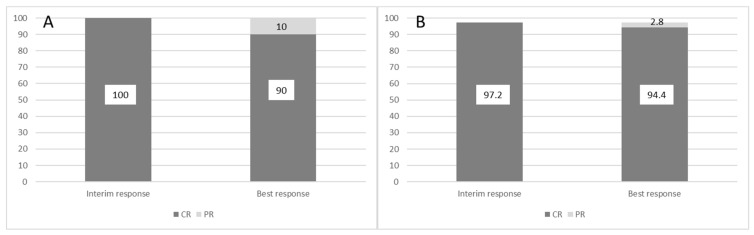
Response evolution between interim evaluation and best-achieved results in patients treated frontline (panel (**A**)) and in those receiving ibrutinib as a second line or beyond (panel (**B**)). Y-axis represents patients %.

**Figure 2 jcm-10-05845-f002:**
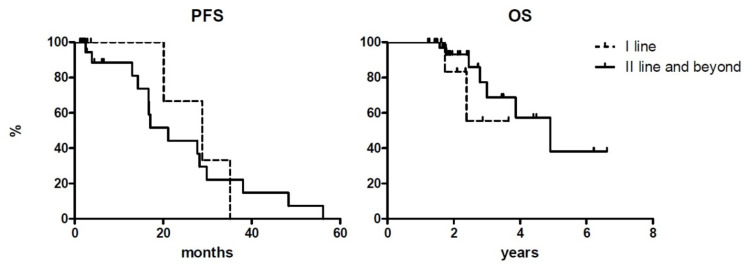
Progression-free survival (PFS) and overall survival (OS) curves according to treatment lines. Y-axis represents patients % of survival.

**Figure 3 jcm-10-05845-f003:**
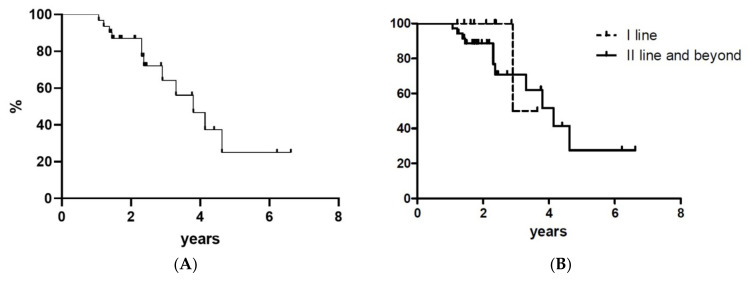
Time-to-next treatment for all treated patients (panel (**A**)) and according to treatment line (panel (**B**)). Y-axis represents patients %.

**Table 1 jcm-10-05845-t001:** Patient characteristics according to treatment line.

	Patients Treated Frontline (*n* = 10)	Patients Treated beyond First Line (*n* = 36)
Median age at diagnosis, years (range)	63.2 (54.1–76.1)	61.6 (32.6–79.2)
Male/female, *n*	8/2	23/13
Previous treatments, median (range)	NA	1 (1–4)
Binet A, *n* (%)	4 (40.0%)	9 (25.0%)
Binet B, *n* (%)	5 (50.0%)	14 (38.9%)
Binet C, *n* (%)	1 (10.0%)	10 (27.8%)
Unavailable, *n* (%)	0	3 (8.3%)
CIRS, median (range)	5 (1–8)	5 (0–16)
*TP53*^mut^, *n* (%)	4 (40.0%)	1 (2.8%)
del(17p), *n* (%)	6 (60.0%)	5 (13.9%)
*IGHV*^unmut^, *n* (%)	5 (50.0%)	12 (33.3%)

NA, not applicable; CIRS, cumulative illness rating scale.

**Table 2 jcm-10-05845-t002:** Outcomes according to treatment line.

	Patients Treated Frontline (*n* = 10)	Patients Treated beyond First Line (*n* = 36)
Best response		
- complete response	1 (10.0%)	1 (2.8%)
- partial response	9 (90.0%)	34 (94.4%)
PFS, median (months)	28.8	21.1
OS, median (years)	NR	4.9

NR, not reached; PFS, progression-free survival; OS, overall survival.

**Table 3 jcm-10-05845-t003:** Hematological adverse events (*).

Toxicity	Any Grade, *n* (%)	Grade ≥ 3, *n* (%)	Drug Correlation, *n* (%)	Dose Reduction, *n* (%)	Resolved, *n* (%)
Neutropenia	10 (16.7)	10 (16.7)	10 (16.7)	6 (10.0)	9 (15.0)
Piastrinopenia	4 (6.7)	2 (3.3)	3 (5.0)	2 (3.3)	3 (5.0)
Anemia	3 (5.0)	3 (5.0)	3 (5.0)	-	3 (5.0)

(*) % were calculated on the total of AEs.

**Table 4 jcm-10-05845-t004:** Comparisons with previously published real-life experience in patients with relapsed and refractory CLL treated with ibrutinib.

	Ibrutinib as 2nd Line Onward				
	This Study	Winqvist 2016 (14)	Ysebaert 2017 (15)	Pula 2020 (16)	van der Straten 2020 (17)
N	36	95	428	171	155
Median age	62	69	70	64	70
*TP53* ^mut^	3%	63% (*)	45% (*)	NA	6%
del(17p)	14%			25%	17%
*IGHV* ^unmut^	33%	NA	NA	NA	NA
Best ORR	97%	84%	89%	77%	67%
Best CR rate	3% (**)	3%	NA	18%	13%
Follow-up	26 mos	10 mos	3 mos	40 mos	14 mos
PFS	Median 21 mos 87% at 1 year 44% at 2 years	Median NR 77% at 10 mos	NA	Median NR 61% at 4 years	Median NR 73% at 1 year
OS	Median 59 mos 100% at 1 year 92% at 2 years	Median NR 83% at 10 mos	NA	Median NR 65% at 4 years	Median NR 77% at 1 year

(*) data collected together for del(17p) and *TP53*^mut^; (**) strictly negative computed tomography scan and bone marrow biopsy. NA, not reported in the paper or not assessed; CR, complete response; mos, months; NR, not reached; ORR, overall response rate; PFS, progression-free survival; OS, overall survival.

**Table 5 jcm-10-05845-t005:** Comparisons with phase 1–3 prospective trials.

	Frontline Ibrutinib				Ibrutinib as 2nd Line Onward		
	This Study	Byrd 2020 (4)	Burger 2020 (8)	Farooqui 2015 (10)	This Study	Byrd 2020 (4)	Munir 2019 (6)
N	10	31	136	35	36	101	195
Median age	63	71	73	62	62	64	67
*TP53* ^mut^	40%	NA	10%	6%	3%	NA	NA
del(17p)	60%	6%	0 (*)	94%	14%	34%	32%
*IGHV* ^unmut^	50%	48%	43%	63%	33%	78%	NA
Best ORR	90%	87%	92%	70%	97%	89%	91%
Best CR rate	10%	35%	30%	12%	3%	10%	11%
Follow-up	24 mos	87 mos	60 mos	24 mos	26 mos	82 mos	65 mos
Median PFS	29 mos	NR	NR	NR	21 mos	52 mos	44 mos
Median OS	NR	NR	NR	NR	59 mos	92 mos	68 mos

(*) patients with del(17p) were excluded from the trial. CR, complete response; mos, months; NA, not reported in the paper; NR, not reached; ORR, overall response rate; PFS, progression-free survival; OS, overall survival.

## Data Availability

The data that support the findings of this study are available from the corresponding author upon reasonable request.

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
