# Peer review of "Long-Term Efficacy and Safety of Ibrutinib in the Treatment of CLL Patients: A Real Life Experience"

_jcm, 2021, doi:10.3390/jcm10245845_

Round 1

Reviewer 1 Report

The manuscript presents a numerically limited experience of patients treated with ibrutinib in a single center in a real-life setting.

The greatest limit of the work is certainly the number of patients, much lower than other real-life series.

Furthermore, some of patients’ characteristics differ from the population commonly seen in real-life, starting from the age that is very low (moreover it is not clear whether age is that of CLL diagnosis, as reported in table 1, or the age at ibrutinib start as per tab 4 and 5).

Patients' PFS is very short and inconsistent with those present in the literature (both in trials and in real-life) even with respect to other series specifically selected for TP53 destruction.

The authors correlate short PFS to unfavorable population characteristics, but in pretreated patients, the median of previous lines is only 1 (it was 3 in RESONATE trial); 25% had Binet stage A; only 33% had unmutated IGHV and less than 20% had TP53 destruction (in this regard, the total number of pts with TP53 destruction would be more clear presenting together TP53 and/or 17p-).

Even in front-line patients, PFS is very low, thus even considering the presence of 40% of patients with TP53 destruction (also here 40% had a stage A of Binet). Even in treatment-naïve the rate of unmutated IGHV is not as high and, anyway, IGHV mutational status is not a risk factor with ibrutinib.

No significant toxicity is reported, therefore not even a possible high incidence or aptitude for ibrutinib  interruptions / discontinuations justifies such reduced PFS.

Other minor considerations:

The 11q patient data is missing;

I would have detailed the characteristics of patients with Richter evolution, especially those on the front line;

The authors hypothesize the correlation between worst outcome in real life and comorbidities of patients (whom, in any case, present a CIRS score<6, also consistent with the median age of both population); in this regard I would add in the discussion 2 experiences focused on the impact of comorbidities with ibrutinib (Gordon et al. Cancer; Tedeschi et al. Blood Adv);

Authors should mention in the discussion the impact of next generation BTK inhibitors that demonstrated comparable efficacy and lower toxicity compared to ibrutinib.

Author Response

#1

The manuscript presents a numerically limited experience of patients treated with ibrutinib in a single center in a real-life setting.

The greatest limit of the work is certainly the number of patients, much lower than other real-life series.

Many thanks for your comment, this limitation was already acknowledged in the Discussion section.

Furthermore, some of patients’ characteristics differ from the population commonly seen in real-life, starting from the age that is very low (moreover it is not clear whether age is that of CLL diagnosis, as reported in table 1, or the age at ibrutinib start as per tab 4 and 5).

This point that you rightly stressed labels the importance of observational study in real-life setting bringing new data for the deepening of the diseases. As reported in Results, 3.1 age at diagnosis was 62 years, highlighted in the revised text for your convenience.

Patients' PFS is very short and inconsistent with those present in the literature (both in trials and in real-life) even with respect to other series specifically selected for TP53 destruction.

The authors correlate short PFS to unfavorable population characteristics, but in pretreated patients, the median of previous lines is only 1 (it was 3 in RESONATE trial); 25% had Binet stage A; only 33% had unmutated IGHV and less than 20% had TP53 destruction (in this regard, the total number of pts with TP53 destruction would be more clear presenting together TP53 and/or 17p-).

Even in front-line patients, PFS is very low, thus even considering the presence of 40% of patients with TP53 destruction (also here 40% had a stage A of Binet). Even in treatment-naïve the rate of unmutated IGHV is not as high and, anyway, IGHV mutational status is not a risk factor with ibrutinib.

No significant toxicity is reported, therefore not even a possible high incidence or aptitude for ibrutinib  interruptions / discontinuations justifies such reduced PFS.

For patients who received ibrutinib as first-line treatment the follow up was shorter than patients who receive ibrutinib after relapse since they were all treated in 2019 whereas patients who received ibrutinib after relapse enter into the study already in 2016. PFS at 1, 2 and 3 years for patients receiving ibrutinib as a salvage treatment was 87.2%, 44.3% and 22.1%, respectively. As already acknowledged, maybe the small sample size could have biased the survival estimation. Nevertheless, as previous stated, we have reported real-life data to offer new information about a rare disease. In addition, we reported a monocentric experience of a referral hospital in which more than 200 trials are ongoing: this could explain a shorter TNT as we have more available new drugs (a new sentence was added in the Discussion section).

Other minor considerations:

The 11q patient data is missing;

The 11q patient data were not available. Nevertheless, many thanks for your suggestion, we’ll take it in consideration for further investigations.

I would have detailed the characteristics of patients with Richter evolution, especially those on the front line;

Only one patient underwent ibrutinib as frontline treatment and deceased due to PD without any other therapy. As they were 5 subjects, we did not believe that these data could represent a useful information. To note, it could be a good hint for further research.

The authors hypothesize the correlation between worst outcome in real life and comorbidities of patients (whom, in any case, present a CIRS score<6, also consistent with the median age of both population); in this regard I would add in the discussion 2 experiences focused on the impact of comorbidities with ibrutinib (Gordon et al. Cancer; Tedeschi et al. Blood Adv);

In total agreement with you we add these two additional references [#21 and 22 in the revised text, highlighted]. References list was shifted accordingly.

Authors should mention in the discussion the impact of next generation BTK inhibitors that demonstrated comparable efficacy and lower toxicity compared to ibrutinib.

Next generation BTK inhibitors are now mentioned in the Discussion, although they are not yet approved for everyday clinical practice in Italy and we cannot yet compare the drugs for both efficacy and safety.

Reviewer 2 Report

Broccoli et al show their results on the use of ibrutinib in their hands. This is an interesting manuscript. Nonetheless there are several questions that should be addressed, some of them major mistakes.

  • Introduction:
    • The authors say: “Moreover, ibrutinib seems capable to overcome the negative impact conferred by high-risk genomic features, such as the deletion of the short arm of chromosome 17 [del(17p)] and the mutation of TP53 (TP53mut)”. What about 11q deletion?
    • Given the fact that the side effects have been observed in the study (results section) and discussed, this subject should be also addressed in the introduction.
    • In the last paragraph “the” is missing (the aim of this study).
  • Materials and methods
    • Why “The diagnosis of CLL had to be established by peripheral blood flow cytometry”?
    • The authors should state how many patients are included in the study, as they are the patients of the study, not the result. Thus, table 1 should be also part of the “materials” section too.
  • Results
    • Table 1 and 2: please, include the number (“n”) in the title of the columns, not as a row. Example: Patients treated frontline, n=10.
    • The authors state that “Ibrutinib has demonstrated a significant clinical impact in patients with de novo and relapsed/refractory chronic lymphocytic leukemia, even in cases with unfavorable cytogenetics and molecular markers”. However, the cytogenetic aberrations are not assessed in all the patients of the cohort, as only 16 out of the 46 CLL patients have cytogenetic information. This is a flaw of the study that should be solved.
    • Regarding the IGHV mutational status, do the authors have this data in the whole cohort or just a subset? If this is the case, it should be commented and considered a limitation of the study.
    • All the statistical analysis fail to show the statistical significance (the value of“p”). This is a major limitation of the study, as it prevent the reader from knowing if the results are relevant.
    • The figure legends should be at the botton of the figure instead of being at the top.
    • The paper could be improved with more elaborated and detailed table explanations and figure legends, they are too short and does not seem to have been done with care/time.
  • Discussion
    • Generally speaking, the purpose of the discussion is to review the study findings in light of the published literature and draw conclusions from the data. However, in this manuscript the comparison with other findings is just done by tables (table 4 nd 5), but not in the text. What are the thought of the authors about these published studies?
    • The results (figures and tables) should be referenced in the text.
    • The study limitations are not considered and objectively discussed Example: the authors say “Moreover, our patients had TP53mut in 40% of cases and an 5 IGHVunmut status in 50%”, however was this analysis done in all the patients?.
  • In general, the paper would benefit from proofreading.

Author Response

#2

Broccoli et al show their results on the use of ibrutinib in their hands. This is an interesting manuscript. Nonetheless there are several questions that should be addressed, some of them major mistakes.

  • Introduction:
  • The authors say: “Moreover, ibrutinib seems capable to overcome the negative impact conferred by high-risk genomic features, such as the deletion of the short arm of chromosome 17 [del(17p)] and the mutation of TP53 (TP53mut)”. What about 11q deletion?

The 11q patient data were not available. Nevertheless, many thanks for your suggestion, we’ll take it in consideration for further investigations.

Given the fact that the side effects have been observed in the study (results section) and discussed, this subject should be also addressed in the introduction.

A sentence was added in the Introduction section as per your right observation (highlighted in the revised text).

o          In the last paragraph “the” is missing (the aim of this study).

We apologize for the typo, “the” was added (highlighted in revised the text).

  • Materials and methods

o          Why “The diagnosis of CLL had to be established by peripheral blood flow

cytometry”?

Peripheral blood flow cytometry is a proof of clonality. This is required for the diagnosis (see Hallek, Blood 2018; 131)

o          The authors should state how many patients are included in the study, as they are the patients of the study, not the result. Thus, table 1 should be also part of the “materials” section too.

A retrospective study was conducted, reviewing our single-centre database and all consecutive patients with a diagnosis of CLL who underwent ibrutinib were eligible. This is the Method. The materials and methods section is used to describe the experimental design and provide sufficient details so that a colleague can repeat the experiment. How many patients we found and analyzed along with their characteristics are the Results we found.

  • Results
  • Table 1 and 2: please, include the number (“n”) in the title of the columns, not as a row. Example: Patients treated frontline, n=10.
  •  

Done.

  • The authors state that “Ibrutinib has demonstrated a significant clinical impact in patients with de novo and relapsed/refractory chronic lymphocytic leukemia, even in cases with unfavorable cytogenetics and molecular markers”. However, the cytogenetic aberrations are not assessed in all the patients of the cohort, as only 16 out of the 46 CLL patients have cytogenetic information. This is a flaw of the study that should be solved.

We are sorry if the information reported in table 1 may lead to misinterpretation. Cytogenetic information is available in all patients. We have 4/10 patients receiving frontline ibrutinib and 1/36 patients receiving ibrutinib beyond first line who have a TP53 mutation (all the others have a wild type TP53). Likewise, del17p was found in 6/10 and 5/36 patients in each subgroup, respectively. All the others have normal cytogenetics.

  • Regarding the IGHV mutational status, do the authors have this data in the whole cohort or just a subset? If this is the case, it should be commented and considered a limitation of the study.

We have the IGHV mutational status for the whole cohort.

  • All the statistical analysis fail to show the statistical significance (the value of“p”). This is a major limitation of the study, as it prevent the reader from knowing if the results are relevant.

Many thanks for your right observation. P values are now reported in the revised paper.

  • The figure legends should be at the botton of the figure instead of being at the top.

Figure legends were moved at the bottom of the figures.

  • The paper could be improved with more elaborated and detailed table explanations and figure legends, they are too short and does not seem to have been done with care/time.

Full explanations are in the text, following Journal’s guidelines legends have to be brief.

  • Discussion
  • Generally speaking, the purpose of the discussion is to review the study findings in light of the published literature and draw conclusions from the data. However, in this manuscript the comparison with other findings is just done by tables (table 4 nd 5), but not in the text. What are the thought of the authors about these published studies?

Comparisons with and comments on other studies are reported in the main text (Discussion section).

  • The results (figures and tables) should be referenced in the text.

Callouts of Tables and Figures are in the main text.

  • The study limitations are not considered and objectively discussed Example: the authors say “Moreover, our patients had TP53mut in 40% of cases and an 5 IGHVunmut status in 50%”, however was this analysis done in all the patients?.

We have the IGHV mutational status for the whole cohort. Limitations and strengths of the study are reported in the Discussion section.

  • In general, the paper would benefit from proofreading.

Many thanks for the suggestion, we revised all the text.

Reviewer 3 Report

I appreciate Author's hard work. Major issues with paper is small patient data. If this is retrospective, we need to increase the "n" to have more power. Ibrutinib is approved for CLL for almost 5yr or so and we have innumerable data on it already. Though authors have a long term follow up, but we have more recent papers with more strong data and hence I donot think it would personally interests readers. Check papers for reference PMID: 34865212,  PMID: 28073846. 

Author Response

In total agreement with you all study limitations are disclosed in the Discussion. The paper was found of interest for the readers by the Guest Editor as it is a monocentric experience of an Italian referral center.

Reviewer 4 Report

Overall the paper is well described even if the quality of the figures is poor.

Please add the y-axis title to figure 1A-B and in figure 2A-B and 3A-B.

Author Response

Many thanks for your comments. Y-axis title is present in the Figures, for more clearness we add a specification in figure legends (highlighted in the text).

Round 2

Reviewer 1 Report

No other suggestions

Author Response

Many thanks, no answer due.

Reviewer 2 Report

-According to the first question, 11q data was not available, but that you say that cytogenetic analysis was performed in all patients, which is a contradiction.

- In the rebuttal letter you say that "We have the IGHV mutational status for the whole cohort". However, in the manuscript you state: "Moreover, our patients had TP53mut in 40% of cases and an IGHVunmut status in 50%". This is not the 100% of cases.

- Although some coments have been followed, this is not the case for some others, which I still consider flaws of the study.

Author Response

According to the first question, 11q data was not available, but that you say that cytogenetic analysis was performed in all patients, which is a contradiction

Cytogenetic analysis was performed in all patients to diagnosed them but 11q is not routinely assessed. According to the recently updated iwCLL recommendations, thorough genetic risk stratification in CLL requires FISH analysis complemented by mutational screening for the TP53, IGHV, del(17p). A specification was added in methods (highlighted in the text).

In the rebuttal letter you say that "We have the IGHV mutational status for the whole cohort". However, in the manuscript you state: "Moreover, our patients had TP53mut in 40% of cases and an IGHVunmut status in 50%". This is not the 100% of cases.

Maybe there was a misunderstanding. The 40% of the total study sample was TP53mut (60% TP53unmut) and the 50% of the total study sample was IGHVunmut (50% IGVHmut).

Although some comments have been followed, this is not the case for some others, which I still consider flaws of the study

In total agreement with you all study limitations are disclosed in the Discussion.

This manuscript is a resubmission of an earlier submission. The following is a list of the peer review reports and author responses from that submission.

Round 1

Reviewer 1 Report

The authors present their experience with the use of ibrutinib in 46 real world CLL patients. The authors compare and contrast their results with published clinical trial data. 

Introduction, line 4: I do not understand what is meant by "deepest impact".

Introduction, line 8: "the best available" should be changed to "an available". There is no evidence that ofatumumab is the best available treatment for CLL. 

Section 2.1: Why were scans performed after the completion of the 4th month of treatment and then every 6 months if the patients weren't in a study? Is this your institutional standard?

11% of patients developed Richter's syndrome. This seems like a relatively high percentage. Do the authors have any thoughts about this?

Discussion, second paragraph: "effictiveness" should be "effectiveness"

Reviewer 2 Report

Authors described about efficacy of ibrutinib in real world data.

In figure 2, OS in patients who received ibrutinib as first-line treatment were extremely short, even majority of them were Binet A or B. Why?

Please add safety assessment in table.

Real ibrutinib dosage or concomitant drug usage are meaningful in clinical setting.

Reviewer 3 Report

In the manuscript entitled “Long-term efficacy and safety of ibrutinib in the treatment of CLL patients: a real life experienced” the authors assess the efficiency of Ibrutinib administered in chronic lymphocytic leukemia patients either as a frontline treatment or as a second line treatment. Even though the purpose of this study is highly important, the findings are limited. The number of patients which are included in this study and moreover the patients which are included in different subgroups with different characteristics is extremely limited. Therefore, no safe conclusions can be done with statistical significance.

It is highly recommended to include more  patients in order to improve the statistical significance of the results as well as to include more information about other studies which assess Ibrutinib as  a frontline or a second line treatment strategy.